# Comparison of Lower Extremity Joint Moment and Power Estimated by Markerless and Marker-Based Systems during Treadmill Running

**DOI:** 10.3390/bioengineering9100574

**Published:** 2022-10-19

**Authors:** Hui Tang, Jiahao Pan, Barry Munkasy, Kim Duffy, Li Li

**Affiliations:** 1Georgia Southern University, Statesboro, GA 30458, USA; 2Boise State University, Boise, ID 83725, USA; 3University of Essex, Wivenhoe Park, Colchester CO4 3SQ, UK

**Keywords:** markerless motion capture system, gait analysis, joint moment, joint power

## Abstract

Background: Markerless (ML) motion capture systems have recently become available for biomechanics applications. Evidence has indicated the potential feasibility of using an ML system to analyze lower extremity kinematics. However, no research has examined ML systems’ estimation of the lower extremity joint moments and powers. This study aimed to compare lower extremity joint moments and powers estimated by marker-based (MB) and ML motion capture systems. Methods: Sixteen volunteers ran on a treadmill for 120 s at 3.58 m/s. The kinematic data were simultaneously recorded by 8 infrared cameras and 8 high-resolution video cameras. The force data were recorded via an instrumented treadmill. Results: Greater peak magnitudes for hip extension and flexion moments, knee flexion moment, and ankle plantarflexion moment, along with their joint powers, were observed in the ML system compared to an MB system (*p* < 0.0001). For example, greater hip extension (MB: 1.42 ± 0.29 vs. ML: 2.27 ± 0.45) and knee flexion (MB: −0.74 vs. ML: −1.17 nm/kg) moments were observed in the late swing phase. Additionally, the ML system’s estimations resulted in significantly smaller peak magnitudes for knee extension moment, along with the knee production power (*p* < 0.0001). Conclusions: These observations indicate that inconsistent estimates of joint center position and segment center of mass between the two systems may cause differences in the lower extremity joint moments and powers. However, with the progression of pose estimation in the markerless system, future applications can be promising.

## 1. Introduction

Inverse dynamics analysis is a fundamental tool widely used for biomechanical studies to understand human movement. The inverse dynamics method combines kinematic and kinetic data with anthropometric parameters and can estimate joint moments and powers [1]. The evaluation of joint moments and powers is critical in clinical decision-making, such as gait retraining [2], treatment with insoles or orthoses [3], and even surgery [4]. Despite its widespread use, the inaccuracy of inverse dynamic analysis stemming from kinematic/kinetic/anthropometric data is well-recognized [5]. Currently, most kinematic data are provided by marker-based (MB) motion-capture systems [6]. However, inaccuracies derived from marker placements, including the center of mass locations [7,8], joint centers [9], the noise due to surface marker movement [10], and skin artifacts [11,12], can be significant barriers. In addition, using MB requires highly trained personnel to avoid human errors when placing markers on participants [13,14]. The intensive time commitment for marker placements and a controlled environment [15] also contribute to the drawbacks of MB. These all make the applications of MB systems challenging in clinical settings with clinical populations [16].

As a critical advancement in vision-based motion capture, a markerless (ML) system offers an alternative approach to measuring kinematic data. Studies have shown the applicability of the deep learning algorithm-based markerless system in gait analysis. Kanko et al. reported excellent agreement with the MB system on spatial parameters (e.g., step length, stride width, and gait speed) and slight differences in temporal parameters (swing time and double support time) [17]. In two follow-up studies, they assessed the lower extremity joint center positions and joint angles. One study emphasized the inter-session variability of joint angles. They reported that the average inter-session variability across all joint angles was 2.8° in the ML system, which is less than all previously reported values (3.0–3.6°) for the MB system [18]. Their other study presented the average systematic root-mean-square joint center differences of 2.5 cm except for the hip joint, which was 3.6 cm. The average systematic root-mean-square for all segment angle differences was 5.5°, except for the rotation angle about the longitudinal axis [19]. These strong results are approaching or superior to the accuracy of MB systems. However, we did not find any lower extremity joint moments and powers in comparison between the two types of systems in the literature. Given the fundamental quantities of interest in human motion research are the intersegmental moments acting at the joints [20], it is critical to compare joint moments estimated by the ML and MB systems. 

Due to marker-based systems’ weakness, markerless systems might introduce new possibilities for inverse dynamic analysis. Therefore, the purpose of the current study was to compare inverse dynamic outcomes of lower extremity joint moments and powers based on ML and MB motion capture systems.

## 2. Materials and Methods

### 2.1. Participants

Recreationally active young adults were recruited for the current study. Inclusion criteria were: (1) free of musculoskeletal injuries and operations of the lower extremity at least 6 months before the data collection and (2) experience with treadmill running. Participants were asked to be free from any intensive exercise within the 24 h before data collection. Participants signed consent forms approved by the Ethics Committee of Georgia Southern University (Approval Number: H22327) before data collection.

### 2.2. Experimental Setup and Procedure

Two camera systems were used in the motion capture procedure: 8 infrared cameras (Vicon Bonita, Oxford, UK) for the MB system to record marker trajectories; and 8 high-resolution video cameras (Vicon Vue, Oxford, UK) for the ML system to record movements. The resolutions of Bonita and Vue cameras are 1 megapixel (1024 × 1024) and 2.1 megapixels (1920 × 1080). Cameras were aimed at the instrumented treadmill (AMTI force-sensing tandem treadmill, Watertown, MA, USA) within a 15.5 m long by 7.6 m wide by 2.4 m tall laboratory space. Camera systems and the instrumented treadmill were synchronized using Vicon Lock+ (Vicon, Oxford, UK), where kinematics were recorded at 100 Hz, and the ground reaction forces were recorded at 1000 Hz.

Before data collection, the cameras were calibrated using an Active Wand (V1, Oxford, UK). Calibration for the Bonita cameras of the MB system and the Vue cameras of the ML system included more than 1000 frames of valid wand data and 600 frames of valid wand data, respectively. The tolerance of image error for MB and ML systems was set as 0.2 and 0.4, respectively. The three-marker option, with an origin marker and two markers for the *X*- and *Y*-axis, was used to set the MB system’s global coordinate system (GCS). An MTD-3 device and CalTester software (CalTester, Motion Lab Systems Inc., Baton Rouge, LA, USA) were used to examine the spatial synchronization between the force plates and cameras, following the manufacturer’s recommended protocol.

When participants arrived at the laboratory, each was introduced to the test protocol. Then, each participant changed into tight shirts and shorts provided by the lab and wore their running shoes. The investigator measured their heights and body mass. Five-minute warm-up exercise and familiarization with treadmill running followed. Following the manufacturer’s suggested procedure, twenty-six 14 mm retro-reflective markers were attached to the participant’s anterior superior iliac spine, posterior superior iliac spine, most lateral prominence of the greater trochanter, lateral prominence of the lateral femoral epicondyle, medial prominence of the medial femoral epicondyle, proximal tip of the head of the fibula, anterior border of the tibial tuberosity, lateral prominence of the lateral malleolus, medial prominence of the medial malleolus, dorsal margin of the first, second and fifth metatarsal head, and aspect of the Achilles tendon insertion on the calcaneus at both sides. A static trial for the MB system was recorded in advance while the participants stood on the treadmill with an anatomical posture. Each participant started walking at an initial speed of 1.12 m/s and then gradually transitioned to running at the target speed of 3.58 m/s. The participants ran on the treadmill for 120 s at 3.58 m/s, and the last 30 s of running were recorded for further analysis. 

### 2.3. Data Analysis

#### 2.3.1. Pre-Processing

Raw marker trajectories were interpolated using Woltring gap filling [21] by Nexus (Vicon Nexus, Oxford, UK). Raw markerless video data were pre-processed by Theia3D (Theia3Dv2022.1.0.2309, Theia Markerless, Inc., Kingston, ON, Canada), where the default IK solution was used to estimate the 3D pose [18]. The lower body kinematic chain has six degrees-of-freedom (DOF) at the pelvis, three DOF at the hip, three DOF at the knee, and six DOF at the ankle. The kinematic and ground reaction force data were filtered through Nexus using a low-pass, zero-lag, 4-order Butterworth filter with cut-off frequencies of 10 Hz and 50 Hz [22], respectively. 

#### 2.3.2. Visual3D Analyses

The pre-processed right lower extremity data were further analyzed using Visual3D (Preview v2022.06.02, C-Motion, Inc., Germantown, MD, USA). 

The same Visual3D 6DOF algorithms and IK constraints for segments were adapted for both systems. IK constraints were set as six DOF at the pelvis, three at the hip, three at the knee, and six at the ankle. For the MB data, the human body was modeled by four linked segments (foot, leg, thigh, pelvis), in which a second kinematic-only foot was created as a virtual foot for kinematic estimations [23]. The segment mass estimations were based on Dempster’s regression equation [24], and inertia properties were computed based on segments as geometrical shapes [25]. The hip joint center was estimated using the method proposed by Bell et al. [26]. Still, the knee and ankle joint centers were estimated using midpoints between external landmarks of the corresponding segment. The anatomical coordinate systems of segments were determined from the static calibration trial. The vertical axis was defined in the direction from distal to proximal joint center, while the anterior–posterior axis was defined as being perpendicular to the vertical axis with no mediolateral component. The third axis was the cross product of the vertical and anterior–posterior axis [27]. The model was automatically created for the ML data based on the deep learning algorithm and segment properties such as segment mass, location of the center of mass, and joint center positions were generated accordingly [19].

Resolved into the proximal coordinate system for both MB and ML data, joint angles and kinetic parameters in the sagittal plane were further calculated. The proximal segment was used as the reference when calculating joint moment and power. Internal joint moments and powers were obtained by applying Newton-Euler methods [1,28], where hip and knee extensor and ankle plantar–flexor moments were assigned to be positive. Positive power values indicated energy production through concentric muscular contractions [1].

Force-based gait events were used to identify stride cycles, in which the force threshold was set at 50 N [29]. The stride cycle was defined as two consecutive right heel contacts. The duration of each stride cycle was scaled to 101 data points. 

#### 2.3.3. Discrete Measurements

The dependent variables were extracted from the last 10 strides from both MB and ML systems. Within each stride, various positive and negative peak values (depending on joint action) in the sagittal plane were identified on moment and power profiles of the hip, knee, and ankle joints, and the relative times to the peak values were included. Presented in Figure 1, peak moments of the hip (top panel) were extension moment in the early stance phase (HM_1_), flexion moment in the stance–swing transition phase (HM_2_), and extension moment at the end of the swing phase (HM_3_); for the knee (middle panel), the extension moment in the early stance phase (KM_1_), and flexion moment at the end of the swing phase (KM_2_); for the ankle (bottom panel), extension moment in the stance phase (AM_1_). Presented in Figure 2, peak powers of the hip (top panel) were absorption power in the middle of the stance phase (HP_1_), the production powers in the early swing phase (HP_2_), and at the end of the swing phase (HP_3_); for the knee (middle panel), absorption powers in the early stance phase (KP_1_), in the early swing phase (KP_3_), and at the end of the swing phase (KP_4_), and production power in the middle of the stance phase (KP_2_); for the ankle (bottom panel), absorption power in the early stance phase (AP_1_), and the production power at the end of the stance phase (AP_2_). 

### 2.4. Statistical Analysis

Means and standard deviations of the differences in kinematic parameters were estimated based on individual measurements between systems. All dependent variables were assessed for normality using a one-sample Kolmogorov–Smirnov test (K-S test, α = 0.05). A two-tailed paired *t*-test was employed based on the normally distributed data to test the differences between the two systems. The effect size was assessed using Cohen’s d [30]. An alpha level of 0.05 was used for statistical analysis. SPSS (22.0, IBM Inc.; Chicago, IL, USA) was used to conduct all statistical analyses. The alpha level was adjusted by 30 dependent variables using the Bonferroni correction to reduce the chances of type I error (α = 0.05/30 = 0.0017).

## 3. Results

Sixteen participants (9 males and 7 females) participated. The participants’ ages, body mass, and height were 23.44 ± 2.31 years, 69.72 ± 9.82 kg, and 1.73 ± 0.08 m, respectively.

### 3.1. Lower Extremity Joint Moments and Powers

Ensemble curves of lower extremity sagittal plane moments and powers estimated using MB and ML are presented in Figure 1 and Figure 2, respectively. Scaled (by body mass) peak magnitudes and relative timing to the peak are presented in Table 1 and Table 2, respectively.

Paired *t*-tests used for analysis based the results that confirmed the normality of the outcome variables. Compared to the MB system, the ML system showed significantly greater peak joint moment magnitudes at HM_2_ (ML: −1.73 ± 0.27, MB: −1.38 ± 0.29), HM_3_ (ML: 2.27 ± 0.45, MB: 1.42 ± 0.29), KM_2_ (ML: −1.17 ± 0.24, MB: −0.74 ± 0.13), and AM_1_ (ML: 3.32 ± 0.55, MB: 3.14 ± 0.51), but less peak magnitude at KM_1_ (ML: 1.28 ± 0.32, MB: 1.40 ± 0.42). For the joint powers, significantly less peak magnitudes were at KP_1_ (ML: −4.05 ± 1.79, MB: −5.0 ± 2.77), KP_2_ (ML: 2.64 ± 1.09, MB: 3.15 ± 1.41), but were greater at HP_2_ (ML: 8.07 ± 2.11, MB: 4.29 ± 1.14), HP_3_ (ML: 5.68 ± 2.71, MB: 3.99 ± 2.13), KP_3_ (ML: −5.42 ± 1.61, MB: −3.45 ± 1.29), KP_4_ (ML: −9.65 ± 2.10, MB: −7.15 ± 1.83), AP_1_ (ML: −9.44 ± 1.81, MB: −8.38 ± 2.48), as well as AP_2_ (ML: 18.40 ± 4.91, MB: 16.07 ± 3.60). In addition, the relative timing to the peak was detected to be significantly different between the MB and ML systems. To be specific, the ML system took longer than the MB system to reach the HM_1_ (ML: 6.74 ± 3.40, MB: 5.16 ± 1.27), HM_2_ (ML: 43.59 ± 7.00, MB: 40.26 ± 6.90), HM_3_ (ML: 92.73 ± 3.00, MB: 90.58 ± 3.39), KM_1_ (ML: 13.53 ± 3.89, MB: 12.98 ± 2.18), KM_2_ (ML: 92.19 ± 2.51, MB: 90.93 ± 2.21), HP_1_ (ML: 28.30 ± 3.68, MB: 26.16 ± 4.45), and KP_1_ (ML: 8.73 ± 3.01, MB: 8.19 ± 2.21). Besides, the ML system took less time than the MB system to reach the HP_3_ (ML: 89.63 ± 4.15, MB: 91.23 ± 3.47). See Table 1 and Table 2 for more details.

### 3.2. Lower Extremity Joint Center and Segment Center of Mass

Joint moments could be significantly affected by joint center position and the segment center of mass. Figure 3 and Figure 4 demonstrate the ensemble curves of differences in joint center positions (hip, knee, ankle) and segment center of mass (thigh, leg, foot) between MB and ML.

In the mediolateral direction (Figure 3 top panel), the ankle (left) and knee (middle) joint centers were biased toward the lateral direction in the ML than the MB throughout the stride cycle. The hip joint center showed the same trend except during initial contact and the late swing phase. In the anterior–posterior direction (Figure 3 middle panel), ML showed a posterior-biased hip joint center during the stride cycle, whereas the ankle and knee joint centers varied within the stride cycle. The ML was posteriorly biased compared to the MB at initial contact for both the ankle and knee joints. For the rest of the stance phase, the ML for the ankle joint was slightly more posterior, and the knee was more anterior. While the ML for the ankle joint continued in the posterior direction in the swing phase, the ML knee continued in the anterior direction in the early swing phase but turned to the posterior direction for the rest of the swing phase. In the vertical direction (Figure 3 bottom panel), the ML of the ankle varied during the stride cycle. In the early stance and mid-swing phase, the estimated bias was toward the superior direction, while in the mid-stance and early swing, it turned to the inferior direction. The ML showed an inferior-biased knee and hip joint center in the early stance. When the ML knee joint turned to the superior direction for the rest of the stride cycle, the hip joint was superior in the mid-stance and early swing phase but moved toward the inferior direction in the rest of the swing phase. See Figure 3 for more details.

In the mediolateral direction (Figure 4 top panel), the center of mass of the foot, leg, and thigh was more lateral in the ML than in the MB systems throughout the stride cycle. In the anterior–posterior direction (Figure 4 middle panel), the foot center of mass was more anterior in the ML than in the MB system during the stride cycle. For the leg center of mass, ML showed more posterior biases than the MB during the stance and swing phases except in different directions at the end of the swing phase. The thigh center of mass was mainly posterior throughout the stride cycle but briefly anterior in the swing–stance transition phase. In the vertical direction, the foot center of mass showed a higher position in the ML than in the MB system during most stance and late swing phases but was lower during the stance–swing transition and early swing phase. For the leg center of mass, the ML demonstrated lower values than the MB system during about 85% of the stride cycle but they were briefly higher during the early swing phase. The thigh center of mass from the ML showed a higher position than the MB system over the whole stride cycle.

### 3.3. Lower Extremity Joint Angles

Ensemble curves of the difference in lower extremity joint angles (hip, knee, ankle joints) in the sagittal plane between the systems are illustrated in Figure 5. 

The hip (top panel) and knee (middle panel) joint angles were biased toward the extension in the ML than MB throughout the stride cycle except briefly for the early swing phase for the hip joint, and early stance phase for the knee joint. On the other hand, ML showed a dorsiflexion-biased ankle joint angle (bottom panel) throughout the stride cycle. 

## 4. Discussion

This study compared MB and ML systems with estimated lower extremity joint moments and powers during treadmill running. Significant differences were detected in the peak magnitudes for joint moments and powers and relative timings to peak estimated by the two systems. Greater peak magnitudes for hip extension and flexion moments, knee flexion moments, and ankle plantarflexion moments, along with their joint powers, were observed in the ML system. Meanwhile, significantly smaller peak magnitudes for knee extension moments coinciding with knee production power were observed.

We focused on the sagittal plane’s joint angles, moments, and powers, since running is primarily a sagittal plane movement [31]. We observed greater hip and knee flexion angles and smaller ankle dorsiflexion angles in the MB system than in the ML system. The tendency was partly consistent with Kanko et al.’s study [19] (Figure 2 for lower extremity joint center, Figure 4 for segment angle and Figure 5 for joint angle), showing that the MB system’s estimation resulted in greater flexion in all three joints. One possible explanation may be the model of the virtual foot. Visual3D introduces three methods to build the virtual foot, which may affect the ankle joint angles. When we chose the heel and toe targets to define the proximal and distal ends of the foot, there was no disclosure for Kanko et al.’s foot model. Different from Kanko et al.’s results, we observed larger magnitudes of systematic differences in the knee and ankle joint angles but smaller magnitudes in the hip joint. Compared to walking in Kanko et al., running in the current project is related to greater lower extremity joint motion [32]. With greater joint motion, soft tissue artifacts can also be larger, leading to an additional 3° error in the joint angles. Additionally, inconsistent marker placement can contribute to a 5° error in the joint angles [14,33].

Compared to the MB system, the ML system’s estimation resulted in greater magnitudes of peak hip flexion moment in the stance–swing transition phase and extension moment in the late swing phase, knee flexion moment in the late swing phase, and ankle plantarflexion moments in the early stance phase, but smaller knee extension moments in the early stance phase. Previous studies presented similar patterns of lower extremity joint moments during running estimated by MB systems. Schache et al. (2011) and Fukuchi et al. (2017) reported lower extremity joint moments at the speed of 3.5 m/s during overground and treadmill running, respectively [34,35]. Their results showed similar peak magnitudes of hip flexion and extension moments of −1.09 and 0.91 Nm/kg (overground) and −1.15 and 1.37 Nm/kg (treadmill); knee flexion moments of −0.53 Nm/kg (overground); as well as the ankle plantarflexion moment with the values of 2.94 Nm/kg (overground) and 2.23 Nm/kg (treadmill). Besides, they reported similar knee extension moments with the values of 3.12 Nm/kg (overground) and 3.18 Nm/kg (treadmill). Consequently, joint powers also showed the same tendency. Despite the differences between overground and treadmill running, Schache et al.’s results revealed similar systematic differences to the present study. Compared to the MB system, the estimations from the ML system may result in greater hip moments and powers in the stance–swing transition and swing phase, knee moments and powers in the swing phase, as well as the ankle moments in the early stance phase, but less knee moments and power in the early stance phase.

The anthropometric model affects the results of the joint moments and powers. Once rigid body equations are set, the joint centers and moment of inertia about the center of mass eventually govern the relationships between kinetics and kinematics [20]. The systematic differences can be further explained by the variations of joint centers and segment mass centers. 

It has been well-recognized that differences in hip joint center location can propagate to hip and knee kinematic and kinetic quantities, especially the hip moments concerning flexion/extension [36]. Besides, the propagation of flexion/extension moments is particularly sensitive to the anteroposterior hip joint center location. We observed that hip joint centers in the ML system were about 2 cm posterior to the MB system in the stance–swing transition and late swing phase. In addition, the sensitivity of hip moments to inertial property variations can be up to 40% [7]. Supported by our results, the thigh center of mass locations was about 2 cm superior in the ML than the MB system in the stance–swing transition phase. Similar to a previous study, greater changes occurred in the swing phase [7], and the ML system showed a biased posterior (about 2 cm) and anterior (about 1 cm) center of mass. For the knee joint, joint moments are sensitive to the differences in knee joint center locations [37]. Previous studies have reported that tibia surface movements affected the knee joint center by 1.1 cm and resulted in the most prominent joint moment in the stance phase [37]. Our results exhibited that knee joint centers differed around 1 cm in all directions and result in greater disparities in the leg center of mass in the early stance phase, which may explain the greater knee extension moment in the MB system. Moreover, our results exhibited a greater leg center of mass in the late swing phase, which may induce greater knee flexion moments. For the ankle joint, we observed less than 1 cm differences in all three directions. Previous studies indicated that average ankle joint position differences were less than 1 cm in the anteroposterior and mediolateral direction and around 2 cm in the vertical direction [19]. Plus, the foot center of mass locations varied greatly. The ML system was slightly biased in the lateral direction (<1 cm) with greater bias in the anterior and superior directions (almost 3 cm). Such a difference may be induced by the marker placements of the first and fifth metatarsal heads. While the MB system reads the markers on the side of the first and fifth metatarsal heads, the markerless system might locate the foot center of mass based on the contour of the shoe. The different joint centers could affect moment arms, and the segment center of mass could affect the estimation of the moment of inertia. Together, they could lead to greater differences in the joint moments and powers.

The methodology differences between ML and MB systems that determine the estimation of poses (body segment positions and orientations) need further attention. To ensure the consistency of the computational algorithms, Visual3D has been used for segments and inverse dynamic solutions used for both systems. However, the MB system depends heavily on physical marker placements over external/internal anatomical landmarks (hardware), and the ML system relies on deep learning-based algorithms to estimate joint center locations (software). The deep learning pose estimation algorithms learn to identify joint centers from the training data. ML in the current study includes over 500,000 manually labeled digital images of the human body and employs biomechanically applicable training data that can identify 51 salient features of the human body [17,18,19]. The optimization methods examine the distance between the manually labeled training data and the estimated joint centers to reduce errors. This process is repeated with the entire training data until improvements between each iteration become negligible. The pose estimation algorithm is then tested on the new image and compared to the training dataset [38]. However, any omissions or biases implicit within training datasets could propagate to situations where the training was weak [19]. Note that the inverse dynamic method is very sensitive to joint center locations, because the estimation of the net joint moments includes the cross product of forces and their moment arms, where the length of moment arms is largely affected by joint center locations [37]. A previous marker-based study has reported that 2 cm superior and 2 cm lateral placement of the hip joint center can decrease the moment of arms of the hip joint by about 28% [38]. Supported by our results, systematic differences in joint center positions have been observed, which might affect moment arms and lead to disparities in joint moments and powers. However, it remains unclear if the differences are caused by marker-based joint center errors induced by soft tissue artifacts [11] or propagations from a weak training dataset.

While the ML system may still be considered in its infancy, evidence from previous studies demonstrated its potential for clinical applications. Since pose estimation algorithms are not dependent on markers attached to the skin, soft tissue artifact errors and human errors usually induced by the MB systems can be eliminated [39]. Studies also presented that the markerless system can extract new information from old datasets [40,41]. Therefore, the ML system can be beneficial in the streamlined monitoring of changes in disease progression [41], rehabilitation [42], athletic training, and competitive sports [43]. In addition, the Theia3D markerless system has shown strong results in the inter-session joint angles variability during walking with loose clothing conditions [18], which the MB systems cannot realize. More importantly, data collection can be completed in a much shorter time than the MB systems [18]. Such benefits could facilitate data collection in a more convenient area with less effect on people’s gait [44], when time is limited, and they are wearing more comfortable clothing. 

However, the differences observed here could have significant implications. For example, previous studies have shown that the values of hip extension and knee flexion moments in the initial stance and late swing phase were important factors in discussing hamstrings injuries during sprinting [45,46]. The greater hip extension and knee flexion moments observed in the late swing with the ML system could impact the hamstring injury-related discussions. Similar to hamstrings, the biarticular rectus femoris plays an important role in the energy transfer between the hip and knee joints. Greater rectus femoris stress is associated with greater hip flexion and knee extension moments. We have observed, with the ML in comparison with the MB system, greater hip flexion moments in the stance to swing transition phase, greater hip power absorption in the late stance phase, and greater hip joint power production in the early swing phase, which could lead to a greater rectus femoris contraction estimation during the stance–swing transition. A previous study showed that greater rectus femoris contraction could lead to greater patellar tendon tension, a risk factor for patellofemoral pain during running [47]. Thus, joint moment/power estimated by the ML system can lead to different assessments for the risk factors of hamstring injuries, patellofemoral pain, and maybe other relevant discussions compared to the MB system. With different risk factor analyses based on the MB or ML systems, clinicians, coaches, and athletes could arrive at different decisions in their practices. 

The following limitations of the current work should be noted. First, our participant pool was limited to recreationally active young adults. Different population groups may have anatomical deformities, affecting the comparison between the systems. Second, we analyzed treadmill running, where the treadmill settings may constrain the speed. Additionally, the hardware and the marker placements are unique to each lab. Despite our updated Vicon high-resolution video cameras employed for markerless data, the Vicon Bonita series infrared cameras used here do not provide us with the highest resolution available on the market. Lower camera resolution may cause trajectory errors in the identifications of landmarks [48]. Therefore, future studies should attempt to replicate the results in different populations using different speeds and hardware settings.

## 5. Conclusions

This study is the first to compare the inverse dynamic outcomes of lower extremity kinetics estimated by the marker-based and markerless systems. We have observed differences in joint moments and powers between the two systems, which could be partially related to the estimations of joint centers and segment center of mass (pose estimations). Although the accuracy and precision of pose estimations between the two systems require further testing, the strengths of the markerless system are apparent. The significantly less data collection and processing time contribute greatly to a more versatile application. With the progression of pose estimation software, the markerless system can be further employed in clinical biomechanics and sports medicine.

## Figures and Tables

**Figure 1 bioengineering-09-00574-f001:**
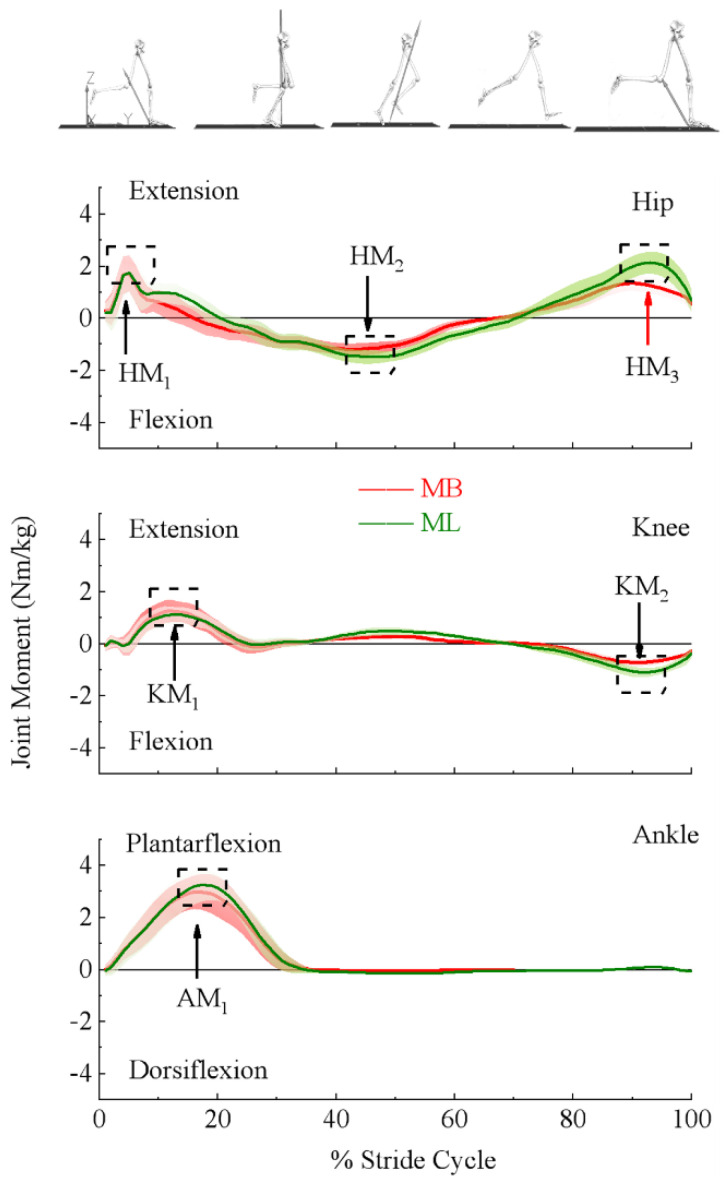
Labeled here are the outcome variables used to quantify the differences in the lower extremity joint moments of the hip (**top panel**), knee (**middle panel**), and ankle (**bottom panel**) (denoted under the ensemble moment curve estimated by marker-based (MB) (red) and markerless (ML) (green) motion capture systems). Joint moments were scaled to participants’ body mass.

**Figure 2 bioengineering-09-00574-f002:**
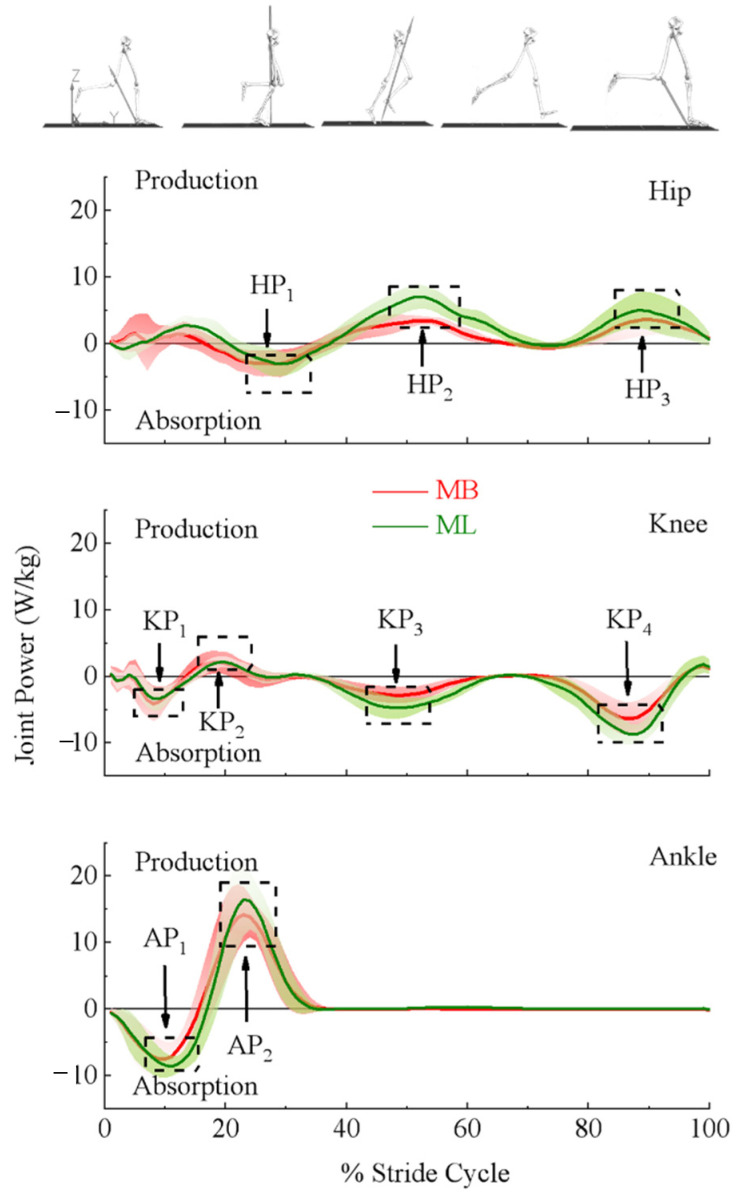
Labeled here are the outcome variables used to quantify the differences in the lower extremity joint powers of the hip (**top panel**), knee (**middle panel**), and ankle (**bottom panel**) (denoted under the ensemble power curve estimated by marker-based (MB) (red) and markerless (ML) (green) motion capture systems). Joint powers were scaled to participants’ body mass.

**Figure 3 bioengineering-09-00574-f003:**
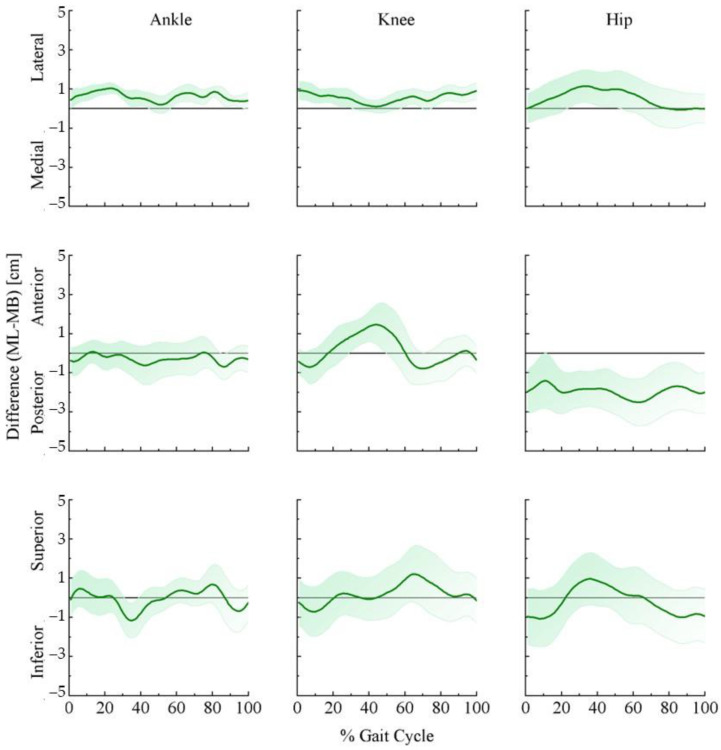
Ensemble curve of lower extremity joint position differences between marker-based and markerless motion capture systems across the average 10 stride cycles for 16 participants. Differences were estimated as markerless (ML) joint center position—marker-based (MB) joint center position.

**Figure 4 bioengineering-09-00574-f004:**
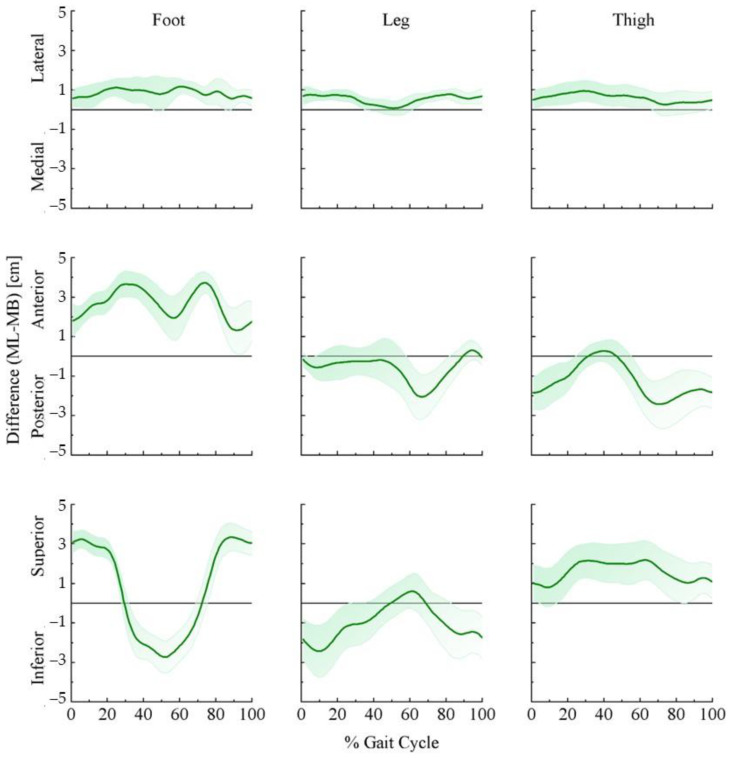
Ensemble curve of lower extremity segment center of mass differences between marker-based and markerless motion capture systems across the average 10 stride cycles for 16 participants. Differences were estimated as markerless (ML) center of mass position—marker-based (MB) center of mass position.

**Figure 5 bioengineering-09-00574-f005:**
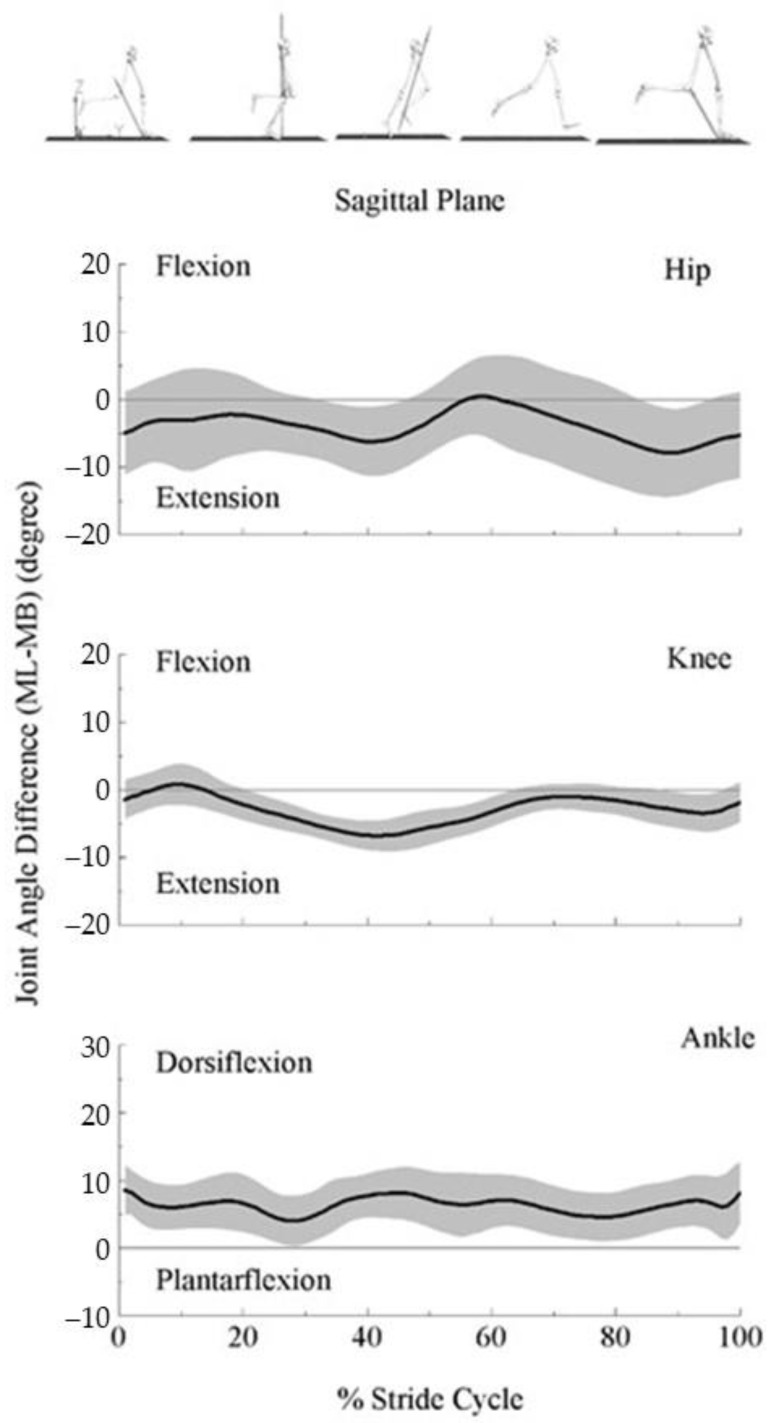
Ensemble curve of lower extremity joint angle differences in the sagittal plane between marker-based (MB) and markerless (ML) motion capture systems across the average 10 stride cycles for 16 participants. Differences were estimated as ML joint angles of each joint—MB joint angles of each joint.

**Table 1 bioengineering-09-00574-t001:** Body mass scaled peak magnitude (Mean, SD) for joint moments and powers for the marker-based (MB) and Markerless (ML) systems.

	Parameters	MB		ML		T & *p*-Value	Cohen’s d
		Mean	SD	Mean	SD		
Moment (Nm/kg)	Hip first peak	2.09	0.66	2.05	0.73	t_159_ = −1.42, *p* < 0.2832	0.1
	*Hip second peak*	* −1.38 *	* 0.23 *	* −1.73 *	* 0.27 *	* t_159_ = −22.99, p < 0.0001 * *	* 1.8 *
	Hip third peak	1.42	0.29	2.27	0.45	t_159_ = 39.612, *p* < 0.0001	3.1
	**Knee first peak**	**1.40**	**0.42**	**1.28**	**0.32**	**t_159_ = 5.907, *p* < 0.0001 ***	**0.5**
	Knee second peak	−0.74	0.13	−1.17	0.24	t_159_ = 40.804, *p* < 0.0001 *	3.2
	**Ankle first peak**	**3.14**	**0.51**	**3.32**	**0.55**	**t_159_ = 10.450, *p* < 0.0001 ***	**0.8**
Power (W/kg)	**Hip first peak**	**−5.02**	**2.60**	**−4.80**	**2.37**	**t_159_ = −1.570, *p* = 0.118**	**0.1**
	Hip second peak	4.29	1.14	8.07	2.11	t_159_ = −27.082, *p* < 0.0001 *	2.1
	Hip third peak	3.99	2.13	5.68	2.71	t_159_ = −13.049, *p* < 0.0001 *	1.0
	**Knee first peak**	**−5.00**	**2.77**	**−4.08**	**1.79**	**t_159_ = −6.138, *p* = 0.0229**	**0.5**
	**Knee second peak**	**3.15**	**1.41**	**2.64**	**1.09**	**t_159_ = 6.628, *p* < 0.0001 ***	**0.5**
	Knee third peak	−3.45	1.29	−5.42	1.61	t_159_ = 21.188, *p* < 0.0001 *	1.7
	Knee forth peak	−7.15	1.83	−9.65	2.10	t_159_ = 31.515, *p* < 0.0001 *	2.5
	**Ankle first peak**	**−8.38**	**2.48**	**−9.44**	**1.81**	**t_159_ = 7.295, *p* = 0.6656**	**0.6**
	**Ankle second peak**	**16.07**	**3.60**	**18.40**	**4.91**	**t_159_ = −10.726, *p* = 0.2426**	**0.9**

K-S tests results for peak moments and peak powers at hip, knee, and ankle joint listed here were all greater than 0.05, therefore normal distribution of the parameters listed in this table was confirmed. * Indicates significant difference; **Bold** indicates the event was observed within the stance phase; *italic with underline* indicates the event was observed during the stance–swing transition; the rest of the parameters were observed within the swing phase. For joint moment data, “+” represents the extension (ankle plantar flexion) and “−” represents the flexion (ankle dorsiflexion). For joint power data, “+” represents energy production, and “−” represents energy absorption.

**Table 2 bioengineering-09-00574-t002:** Relative Time to Peak as Percentage Stride Cycle (Mean, SD) for Joint moments and Powers for Marker-based (MB) and Markerless (ML) Systems.

	Parameters	MB		ML		T & *p*-Value	Cohen’s d
		Mean	SD	Mean	SD		
Moment (%stride Cycle)	**Hip first peak**	**5.16**	**1.27**	**6.74**	**3.40**	**t_159_ = −5.946, *p*< 0.0001 ***	**0.5**
	*Hip second peak*	* 40.26 *	* 6.90 *	* 43.59 *	* 7.00 *	* t_159_ = −7.179, p < 0.0001 * *	* 0.6 *
	Hip third peak	90.58	3.39	92.73	3.00	t_159_ = −8.637, *p* < 0.0001 *	0.7
	**Knee first peak**	**12.98**	**2.18**	**13.53**	**3.89**	**t_159_ = −2.008, *p* = 0.0015 ***	**0.2**
	Knee second peak	90.93	2.21	92.19	2.51	t_159_ = −10.031, *p* < 0.0001 *	0.8
	**Ankle first peak**	**18.29**	**1.90**	**18.33**	**1.82**	**t_159_ = −0.569, *p* = 0.57**	**0.0**
Power (%Stride Cycle)	**Hip first peak**	**26.61**	**4.45**	**28.30**	**3.68**	**t_159_ = −6.983, *p* < 0.0001 ***	**0.6**
	Hip second peak	51.89	3.90	52.36	4.37	t_159_ = −1.243, *p* = 0.216	0.1
	Hip third peak	91.23	3.47	89.63	4.15	t_159_ = 4.458, *p* < 0.0001 *	0.4
	**Knee first peak**	**8.19**	**2.21**	**8.73**	**3.01**	**t_159_ = −2.380, *p* = 0.0006 ***	**0.2**
	**Knee second peak**	**19.45**	**3.09**	**21.17**	**3.51**	**t_159_ = −6.606, *p* < 0.0001 ***	**0.5**
	Knee third peak	49.31	2.77	49.06	3.30	t_159_ = 0.860, *p* = 0.391	0.1
	Knee forth peak	86.59	2.18	86.83	2.31	t_159_ = −1.740, *p* = 0.084	0.1
	**Ankle first peak**	**10.58**	**2.20**	**10.82**	**2.22**	**t_159_ = −1.794, *p* = 0.075**	**0.6**
	**Ankle second peak**	**24.10**	**2.32**	**24.21**	**2.08**	**t_159_ = −0.943, *p* = 0.347**	**0.1**

K-S tests results for relative timing to peak moments and peak powers at hip, knee, and ankle joints listed here were all greater than 0.05, therefore normal distribution of the parameters listed in this table was confirmed. * Indicates significant difference; **Bold** indicates the event was observed within the stance phase; *italic with underline* indicates the event was observed during the stance–swing transition; rest of the parameters were observed within the swing phase.

## Data Availability

Not applicable.

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
