# Peer review of "Comparison of Lower Extremity Joint Moment and Power Estimated by Markerless and Marker-Based Systems during Treadmill Running"

_bioengineering, 2022, doi:10.3390/bioengineering9100574_

Round 1
Reviewer 1 Report
1) I repeatedly find the word "moment" in the article and in the title. Is it "moment" or "movement"?
2) Is it possible to provide any specific problem statement in the abstract? I find the statement "no research is done" very vague.
3) The literature survey section lacks the critical analysis of the existing works.
4) Will there be any change in the results if the resolution of the camera is changed? What are the specifications of the video data generated?
5) It will be better if some screenshots of the 3D video data is shown.
6) Do you foresee any relation between the age and the movements of the users? What if the same experiment is repeated with a different group of people of different age?
7) How do you validate your results?
8) Statistical analysis section seems to be very small. More information must be added here.
9) Will the settings/environment of treadmill affect your results? How?
10) It is not ideal to specify the references in the discussion section since it is your own findings/inferences.
Author Response
Reviewer 1:
- I repeatedly find the word "moment" in the article and in the title. Is it "moment" or "movement"?
Author’s Response: Thank you for the comment. It is “joint moment”, not “movement”. It is the product of force and moment-arm. See the first book we cited on the reference list. This is one of the most popular biomechanics textbooks researchers commonly referred to for methods and terminologies.
- Is it possible to provide any specific problem statement in the abstract? I find the statement "no research is done" very vague.
Author’s Response: Thank you for the comment. We have added relevant statements in the introduction to address this issue (line 38-50) although this is an exploratory, not a hypothesis-driven project.
- The literature survey section lacks the critical analysis of the existing works.
Author’s Response: Thank you for the comment. The introduction section has been rewritten and more analysis of current literature has been added.
- Will there be any change in the results if the resolution of the camera is changed? What are the specifications of the video data generated?
Author’s Response: Thank you for the comment. We believe that a change in camera resolutions would lead to changes in the results. However, the current study chose to adopt the camera resolutions based on manufacturers’ suggestions as well as the commonly used ones in the previous literature. The resolutions of cameras for both Bonita cameras and Vue cameras were adaptable to our study environment. We have added relevant verbiage in the limitation section to reflect your concerns (line 455-61).
- It will be better if some screenshots of the 3D video data is shown.
Author’s Response: We appreciate your suggestion. However, previous studies conducted by Kanko et al. (2021) have presented the visual results of Theia3D in their publications. It may look repetitive although screenshots of 3D video data can provide more vivid results.
6) Do you foresee any relation between the age and the movements of the users? What if the same experiment is repeated with a different group of people of different age?
Author’s Response: Thank you for the comment. Yes, there should be relations between age and movements. Future projects should investigate such relationships using either marker-based or markerless systems.
- How do you validate your results?
Author’s Response: Thank you for the comment. We have listed a few limitations at the end of the manuscript (lines 451-61). Changes in the testing populations, tasks, and testing environments can be used to validate our results.
- Statistical analysis section seems to be very small. More information must be added here.
Author’s Response: Thank you for the comment. It is a simple within-subject design. Paired t-test is the optimal option based on our study purpose.
- Will the settings/environment of treadmill affect your results? How?
Author’s Response: Thank you for the comment. Yes, we believe so. The current environment settings fitted our study design. In addition, we stated the limitations of environment settings in the discussion section (line 455-61highlighted).
10) It is not ideal to specify the references in the discussion section since it is your own findings/inferences.
Author’s Response: Thank you for the comment. We have reworded the discussion section for clarification.
Reviewer 2 Report
1) In the literature survey section, the problems of the existing methods are not clearly explained and the results of the work related to them are not properly explained.
2) Explanation of figures are not clear.
3) In the abstract, the authors should place more results and values on this part. Add more explanation in conclusion part.
4) Give more explanation in existing method about result and discussion part.
5) Some recent reference documents should be included in related journals
6) Novelty is missing .need to explain
7) Add more explanation in conclusion part.
8) Need more explanation in result analysis
9) The Language errors is seen in many places in the manuscript and typo errors are also seen in the manuscript. Revise the manuscript accordingly
Author Response
1). In the literature survey section, the problems of the existing methods are not clearly explained and the results of the work related to them are not properly explained.
Author’s Response: Thank you for the comment. We have highlighted the drawbacks of the marker-based system in the introduction section (line 38-50). In addition, we have added the importance of comparing the kinetic parameters between MB and ML systems (lines 432-50).
2) Explanation of figures are not clear.
Author’s Response: Thank you for the comment. We edited the captions of figures 1 and 2 in the revised manuscript for clarification (line 179 and line 187). Figures 3-5 are for illustration purposes, so we kept the original captions.
3) In the abstract, the authors should place more results and values on this part. Add more explanation in conclusion part.
Author’s Response: Thank you for the comment. We added the values of the key outcome variables in the abstract (line 21-3) but still tried to stay within the character limitations of the abstract,
4) Give more explanation in existing method about result and discussion part.
Author’s Response: Thank you for the comment. Results and discussion sections have been revised and the explanations have been highlighted.
5) Some recent reference documents should be included in related journals
Author’s Response: Thank you for the comment. We include the most recent related references in the current study. For example, we cited the review paper published in 2022 and three papers (2021) investigated the same ML motion capture system.
- Wade, L., et al., Applications and limitations of current markerless motion capture methods for clinical gait biomechanics. PeerJ, 2022. 10: p. e12995.
- Kanko, R.M., et al., Assessment of spatiotemporal gait parameters using a deep learning algorithm-based markerless motion capture system. Journal of Biomechanics, 2021. 122: p. 110414.
- Kanko, R.M., et al., Inter-session repeatability of markerless motion capture gait kinematics. Journal of Biomechanics, 2021. 121: p. 110422.
- Kanko, R.M., et al., Concurrent assessment of gait kinematics using marker-based and markerless motion capture. Journal of Biomechanics, 2021. 127: p. 110665.
6) Novelty is missing .need to explain
Author’s Response: Thank you for the comment. We have edited the introduction to highlight the novelty of the current study (line 66-8).
“Given the fundamental quantities of interest in human motion research are the intersegmental moments acting at the joints [20], it is critical to compare joint moments estimated by the ML and MB systems”.
7) Add more explanation in conclusion part.
Author’s Response: Thank you for the comment. We have edited the conclusion for clarity.
8) Need more explanation in result analysis
Author’s Response: Thank you for the comment. We have edited the results for clarification.
9) The Language errors is seen in many places in the manuscript and typo errors are also seen in the manuscript. Revise the manuscript accordingly
Author’s Response: Thank you for the comment. We have edited the manuscript throughout to improve its readability.
Reviewer 3 Report
This study aimed to compare lower extremity joint moments and powers estimated by marker-based (MB) and ML motion capture systems. The force data were recorded via an instrumented treadmill. Results: Greater peak magnitudes for hip extension and flexion moments, knee flexion moment, and ankle plantarflexion moment, along with their joint powers, were observed in the ML system compared to an MB system.
I value the authors efforts to present their significant research work. It seems to me that the subject matter is presented in a comprehensive manner. Overall, the paper quality largely meets the requirements of "Bioengineering", hence the manuscript could be accepted for publication after revision. 1. Literature review could be more focused. Maybe there lacks detailed explanations about the key contributions. The readers need more help to understand what is important, what is new, and how it relates to the state of art. 2. What are the limitations of the proposed lower extremity joint moments and powers estimation method? The author did not mention it in the paper. What are the implications of the findings? More discussion should be provided in the manuscript. The factors that influence the accuracy should be analyzed in more detail in the discussion section. 3. It would be nice to discuss the following papers in the paper: - Celik, Y., Stuart, S., Woo, W. L., Sejdic, E., & Godfrey, A. Multi-modal gait: A wearable, algorithm and data fusion approach for clinical and free-living assessment. Information Fusion, 2022, 78, 57-70. - S. Qiu, H. Zhao, N. Jiang, Z. Wang, L. Liu, Y. An, H. Zhao, X. Miao, R. Liu, G. Fortino. Multi-sensor information fusion based on machine learning for real applications in human activity recognition: State-of-the-art and research challenges, Information Fusion, 2022, 80:241-265 - Pham, T. T., Suh, Y. S.. Conditional Generative Adversarial Network-based Regression Approach for Walking Distance Estimation using Waist-mounted Inertial Sensors. IEEE Transactions on Instrumentation and Measurement. 2022, 71, 2510113 4. Transitions from section to section should be smoother. 5. Proofread the paper and improve readability.Author Response
This study aimed to compare lower extremity joint moments and powers estimated by marker-based (MB) and ML motion capture systems. The force data were recorded via an instrumented treadmill. Results: Greater peak magnitudes for hip extension and flexion moments, knee flexion moment, and ankle plantarflexion moment, along with their joint powers, were observed in the ML system compared to an MB system.
I value the authors efforts to present their significant research work. It seems to me that the subject matter is presented in a comprehensive manner. Overall, the paper quality largely meets the requirements of "Bioengineering", hence the manuscript could be accepted for publication after revision.
- Literature review could be more focused. Maybe there lacks detailed explanations about the key contributions. The readers need more help to understand what is important, what is new, and how it relates to the state of art.
Author’s Response: Thank you for the comment. We have stressed on the shortcomings of MB systems in the introduction section (line 38-49, 66-8).
“Currently, most kinematic data are provided by marker-based (MB) motion capture systems [6]. However, inaccuracies derived from marker placements, including the center of mass locations [7, 8], joint centers [9], the noise due to surface marker movement [10], and skin artifacts [11, 12], can be significant barriers. In addition, using MB requires highly trained personnel to avoid human errors when placing markers on participants [13, 14]. Intensive time commitment for marker placements, and a controlled environment [15] also contribute to drawbacks of MB. These all make the applications of MB systems challenging in the clinical settings with clinical populations [16].”
- What are the limitations of the proposed lower extremity joint moments and powers estimation method? The author did not mention it in the paper. What are the implications of the findings? More discussion should be provided in the manuscript. The factors that influence the accuracy should be analyzed in more detail in the discussion section.
Author’s Response: Thank you for the comment. The implications had been addressed in the discussion. We have highlighted this part (lines 432-450). The factors that would influence the accuracy include joint center, joint center of mass, and environment settings. The former two had been analyzed and explained in detail in the discussion. The latter one had been listed as limitations.
- It would be nice to discuss the following papers in the paper: - Celik, Y., Stuart, S., Woo, W. L., Sejdic, E., & Godfrey, A. Multi-modal gait: A wearable, algorithm and data fusion approach for clinical and free-living assessment. Information Fusion, 2022, 78, 57-70. - S. Qiu, H. Zhao, N. Jiang, Z. Wang, L. Liu, Y. An, H. Zhao, X. Miao, R. Liu, G. Fortino. Multi-sensor information fusion based on machine learning for real applications in human activity recognition: State-of-the-art and research challenges, Information Fusion, 2022, 80:241-265 - Pham, T. T., Suh, Y. S.. Conditional Generative Adversarial Network-based Regression Approach for Walking Distance Estimation using Waist-mounted Inertial Sensors. IEEE Transactions on Instrumentation and Measurement. 2022, 71, 2510113
Author’s Response: Thank you for the comment. These papers investigated wearable inertia sensors. Although inertia sensors are portable devices that have been widely reported with convenient and reliable for clinicians to collect motion capture data, they are different technologies from our current compared ones. The two main innovations in biomechanics data collection in recent years, wearable and markerless are both discussed in the literature in comparison to the marker-base systems. The comparison between wearable and markerless systems is important for future research projects.
- Transitions from section to section should be smoother.
Author’s Response: Thank you for the comment. We have edited the paper for readability.
- Proofread the paper and improve readability.
Author’s Response: Thank you for the comment. We have edited the paper for readability.
Reviewer 4 Report
In this paper, the authors compared the performance of detecting lower extremity joint moment and power between markerlessl and marker-based Systems. Overall, the paper is well structured. Hoiiivever„ due to the lack of some details, improvements are essential to achieve publishable standards.
1. "...which can be challenging in some clinical settings with 42 clinical populations [15]." The limitations of MB systems need to be specified.
2. "Following standard VICON procedure..." Full name is necessary when the acronym first appears.
3. "Two-176 tailed paired t-test was used to test the differences between the two systems.” Non-parametric test should be performed instead of paired t-test if the data in any group do not satisfy normal distribution.
4. In the results, it was not shown if normal distribution was satisfied.
5. In section 3.1 there are too many numbers. These data have been listed in Table 1. It would be better to summarize them instead of listing in text,
6. "Compared to the MB system, the PAL system's estimation resulted in greater magnitudes of peak hip flexion..." In this paragraph, the values do not need to be repeated.
7. The present study has guaranteed consistency in the computational algorithms for segments and inverse dynamic solutions for both systems in the Visual3D." This is a pilot study with a small scale of data. This is overstated. The consistency need further justification in different body movement status.
8. Due to the difference in body movement and sensor attachment mode bet Teen different body sites (Refer: 10.3389/f phys.2020.00823 10.1007/s10877-020-00481-3)„ it deserves further investigation to see if the conclusions is can be adopted to other measurement sites, different physiological status (e.g.., posture }, and other non-contact body movement detection technologies (e.g.., radar and WIFI, refer: 10.33901s22000809).
Author Response
In this paper, the authors compared the performance of detecting lower extremity joint moment and power between markerlessl and marker-based Systems. Overall, the paper is well structured. Hoiiivever„ due to the lack of some details, improvements are essential to achieve publishable standards.
- which can be challenging in some clinical settings with 42 clinical populations [15]." The limitations of MB systems need to be specified.
Author’s Response: Thank you for the comment. We have edited the limitations of MB systems in the introduction part (line 38-50).
- Following standard VICON procedure..." Full name is necessary when the acronym first appears.
Author’s Response: Thank you for the comment. We have edited the sentence from the manufacture’s name (VICON) to “manufacturer’s suggested procedure” in the revised manuscript (line 106).
- Two-176 tailed paired t-test was used to test the differences between the two systems.” Non-parametric test should be performed instead of paired t-test if the data in any group do not satisfy normal distribution.
Author’s Response: Thank you for the comment. The normality had been reported in the 2.4. Statistical analysis “All dependent variables were assessed for normality using a one-sample Kolmogorov-Smirnov test (α=0.05)”. Relevant verbiage also added to the results section (lines 211-2).
- In the results, itwas not shown if normal distribution was satisfied.
Author’s Response: Thank you for the comment. Relevant verbiage added to the results section (lines 211-2).
- In section 3.1 there are too many numbers. These data have been listed in Table 1. It would be better to summarize them instead of listing in text,
Author’s Response: Thank you for the comment. We presented the mean±standard deviation to show magnitudes differences. Additional statistical results, t-values, degree of freedom involved in the tests, and p-values are all reported in the tables.
- Compared to the MB system, the PAL system's estimation resulted in greater magnitudes of peak hip flexion..." In this paragraph, the values do not need to be repeated. Author’s Response: Thank you for the comment. The numbers have been edited out as suggested, see lines 326-330.
- The present study has guaranteed consistency in the computational algorithms for segments and inverse dynamic solutions for both systems in the Visual3D." This is a pilot study with a small scale of data. This is overstated. The consistency need further justification in different body movement status.
Author’s Response: Thank you for the comment. We have tuned down the statement (Line 385-8): “To ensure the consistency of the computational algorithms, Visual3D has been used for segments and inverse dynamic solutions used for both systems”.
- Due to the difference in body movement and sensor attachment mode bet Teen different body sites (Refer: 10.3389/f phys.2020.00823 10.1007/s10877-020-00481-3)„ it deserves further investigation to see if the conclusions is can be adopted to other measurement sites, different physiological status (e.g.., posture }, and other non-contact body movement detection technologies (e.g.., radar and WIFI, refer: 10.33901s22000809).
Author’s Response: We agree. Further investigations should focus on different measurement sites and statuses (lines 457-9).
Round 2
Reviewer 1 Report
It can be accepted now
Author Response
It can be accepted now.
Authors response: Thank you.
Reviewer 4 Report
Thanks for the update. Some issues mentioned in my earlier comments still need further improvement.
1. Regarding the paired t-test, the authors added "... based the results that confirmed the normality of the 480 outcome variables." However, I suggest to clearly mention the results of K-S test in all subgroups.
2. As mentioned in my earlier comments, the performance need to be comprehensive evaluated in different physiological conditions (body sites, postures, etc.). Since the location (i.e., body site) is a main focus of this paper, statistical results should be added to show if the difference (both original and relative) between MB and ML systems is dependent on the location.
Author Response
Thanks for the update. Some issues mentioned in my earlier comments still need further improvement.
- Regarding the paired t-test, the authors added "... based the results that confirmed the normality of the 480 outcome variables." However, I suggest to clearly mention the results of K-S test in all subgroups.
Authors response: Thank you for your suggestion. We have added relevant information at the bottom of Tables 1 and 2.
- As mentioned in my earlier comments, the performance need to be comprehensive evaluated in different physiological conditions (body sites, postures, etc.). Since the location (i.e., body site) is a main focus of this paper, statistical results should be added to show if the difference (both original and relative) between MB and ML systems is dependent on the location.
Authors response: Thank you for your comment. The focus is on the lower extremity sagittal plane joint moments and power comparisons between the two systems. We have listed all of the T-values, P-values, and Cohen’s D-values in Tables 1 and 2 for all of the body sites that are relevant to this study. However, Kanko et al. (2021) reported the comparisons of lower extremity kinematic variables, lower extremity joint center (Fig. 2), segment angle (Fig.4), and joint angle (Figs 5 and 6). Our kinematic results are comparable to their report. To avoid redundancy, we direct the readers’ attention to their study (lines 310-311), so we can focus on the comparisons of the kinetic variables.